# KARATECH: A Practice Support System Using an Accelerometer to Reduce the Preliminary Actions of Karate

**DOI:** 10.3390/s24072306

**Published:** 2024-04-05

**Authors:** Kwangyun Kim, Shuhei Tsuchida, Tsutomu Terada, Masahiko Tsukamoto

**Affiliations:** 1Graduate School of Engineering, Kobe University, 1-1 Rokkodai-Cho, Nada-Ku, Kobe 657-8501, Hyogo, Japan; kwangyun-kim@stu.kobe-u.ac.jp (K.K.); tuka@kobe-u.ac.jp (M.T.); 2Center for Interdisciplinary AI and Data Science, Ochanomizu University, 2-1-1 Otsuka, Bunkyo-Ku, Tokyo 112-8610, Japan; tsuchida.shuhei@ocha.ac.jp

**Keywords:** human motion analysis, accelerometer, sports support, karate, dynamic time warping

## Abstract

Kumite is a karate sparring competition in which two players face off and perform offensive and defensive techniques. Depending on the players, there may be preliminary actions (hereinafter referred to as “pre-actions”), such as pulling the arms or legs, lowering the shoulders, etc., just before a technique is performed. Since the presence of a pre-action allows the opponent to know the timing of the technique, it is important to reduce pre-actions in order to improve the kumite. However, it is difficult for beginners and intermediate players to accurately identify their pre-actions and to improve them through practice. Therefore, this study aims to construct a practice support system that enables beginners and intermediate players to understand their pre-actions. In this paper, we focus on the forefist punch, one of kumite’s punching techniques. We propose a method to estimate the presence or absence of a pre-action based on the similarity between the acceleration data of an arbitrary forefist punch and a previously prepared dataset consisting of acceleration data of the forefist punch without a pre-action. We found that the proposed method can estimate the presence or absence of a pre-action in an arbitrary forefist punch with an accuracy of 86%. We also developed KARATECH as a system to support the practice of reducing pre-actions using the proposed method. KARATECH shows the presence or absence of pre-actions through videos and graphs. The evaluation results confirmed that the group using KARATECH had a lower pre-action rate.

## 1. Introduction

There are two main types of karate competitions: kata and kumite. Kumite is a sparring competition in which two players face off and perform offensive and defensive techniques. In kumite, the winner is determined by how well the player can attack the opponent and how well the player can cope with the opponent’s attacks. Therefore, it is important to reduce preliminary actions (hereinafter referred to as “pre-actions”) during an attack, such as fist motion, arm lowering, shoulder raising, etc., as these will inform the opponent of the timing of their punch or kick and give the opponent time to prevent the attack. Expert players can know their pre-actions and keep them small. However, it is difficult for beginners and intermediate players to accurately grasp their pre-actions, and it is difficult for them to minimize their pre-actions through self-practice. For beginners and intermediate players to deepen their understanding of pre-actions, it is useful to have a system that automatically detects only the pre-action part of a series of kumite motions and provides easy-to-understand feedback.

In this study, we developed KARATECH, a practice support system that enables beginners and intermediate players to accurately grasp and understand their pre-actions. KARATECH automatically extracts the pre-action part from the measured acceleration data during the forefist punch and shows the presence or absence of pre-actions through videos and graphs. Gesture recognition using acceleration data is generally based on SVM (support vector machine) [1], RF (random forest) [2], HMM (hidden Markov model) [3], and DTW (dynamic time warping) [4], etc.

We had difficulty identifying small gestures buried in large gestures, such as pre-actions, in conventional algorithmic gesture recognition methods [5,6,7,8] because the identification result tends to be determined by large gestures in a series of gestures. In this study, we propose a method for estimating the presence or absence of pre-actions. The proposed method analyzes the waveform of acceleration data in a punch by tracing back the time series from the peak value of the waveform of acceleration data to the beginning of the punch. The proposed method not only detects the presence or absence of a pre-action associated with a punching motion, which has been difficult to recognize, but also obtains the segment containing the pre-action.

The rest of this paper is organized as follows: Section 2 introduces related research, Section 3 explains the proposed method, Section 4 discusses the proposed system, and Section 5 summarizes the paper.

## 2. Related Work

### 2.1. Evaluation of Karate Movements Using Sensors

There are many studies that use various sensors to evaluate karate motions and to assist in practice. Hachaj et al. presented a method for classifying various karate kicks using motion capture to analyze and visualize the 3D trajectory of the lower body joints in human motor activity [9,10]. Dana et al. used DTW to analyze motion capture data of karate kata performed by multiple participants with different timing and velocities [11]. Although motion capture systems can visualize and evaluate detailed motions in karate, it has been pointed out that the use of many markers attached to the body limits motion. They are also expensive, require large equipment, and are limited in space, making them difficult for anyone to use. Ait-Bennacer et al. proposed a karate smart coaching system that applies deep learning and computer vision techniques [12]. The system can classify eight different karate techniques with 96% accuracy. However, the system is limited in space, as is the motion capture system described above, because it requires multiple cameras.

Vencesbrito et al. used EMG (electromyography) to measure arm electromyographic patterns during punches of experienced and inexperienced karate players. With this, they characterized the kinematic and electromyographic patterns of the experienced players’ punches and compared how these patterns differ from those of inexperienced karate players [13]. However, EMG is not suitable for the measurement of intense gestures, and it is difficult to maintain contact between the electrode and the skin.

In this study, we analyze and evaluate karate motions using an accelerometer, which has become smaller and less expensive in recent years and is widely used in the sports field. The accelerometer is less spatially constrained than optical motion capture and less physically constrained than EMG. Therefore, the accelerometer is suitable for measurement in sports practice.

In a study of karate motions using inertial sensors including the accelerometer, Vuković et al. analyzed in detail the punches of advanced players using two inertial sensors in order to highly improve their proficiency in kumite techniques [14]. Yadav et al. proposed MS-KARD, a multi-stream karate motion recognition dataset using two cameras and three inertial sensors [15]. They also proposed KarateNet, a fusion-based motion recognition network trained on MS-KARD. The main purpose of these studies is to classify karate techniques, and there are few studies on the recognition of pre-actions.

### 2.2. Motion Recognition Method Using the Inertial Sensor

Many methods have been studied to recognize actions such as posture, motion, and gestures using the accelerometer, the gyroscope, and the IMU. Murao et al. investigated the effect on gesture recognition accuracy of changing the number and position of sensors and the number and type of gestures [16]. They captured data for 27 kinds of gestures by using a mobile device with nine accelerometers and nine gyroscopes for the investigation. They found the best positioning of the accelerometer and the gyroscope for effective recognition accuracy due to interdependence among gestures. Pernek et al. developed a system to recognize the type and intensity of arm muscle training with a barbell using five IMUs [17]. They used a two-layered SVM to recognize six types of training with an accuracy of 85% and an intensity prediction error of 6%. Otoda et al. developed a sensing chair with six IMUs on the seat and two IMUs on the back of the chair to estimate 18 different postures and the presence or absence of the seated person. A learning model was constructed using RF and classified the sitting postures with an accuracy of 80%. Junker et al. [18] and Georgi et al. [19] proposed a method to recognize specific gestures by HMM and achieved high recognition accuracy. HMM is highly versatile for recognizing various gesture patterns because it models the probability distribution of state transitions and is suitable for modeling long-term time dependence. On the other hand, it lacks ease of implementation because it is difficult to design an appropriate model, requires a large amount of teaching data for parameter estimation, and is computationally expensive.

Like HMM, DTW handles time series information, and computation time increases for long time series data because of the calculation of the distance among all the data. However, since the time required for a pre-action in karate is very short, DTW can keep the computation time down to a practically feasible level. Therefore, we use a recognition method based on DTW in this study. DTW is an algorithm that can measure the similarity of time series data even if they have different lengths and periods. DTW measures the similarity between two time series by brute-force calculation of the distance between each point in the time series data. The karate techniques and pre-actions that are the objects of recognition in this study do not have a fixed time length or speed. Therefore, we believe that DTW is suitable for their recognition.

Many studies have modified DTW to meet their research objectives. Gao et al. proposed a two-dimensional dynamic time warping algorithm (2D-DTW) that can directly measure the similarity between high-dimensional data such as matrices [20]. Xu et al. proposed comparative dynamic time warping (C-DTW) to solve the problem that the accuracy of real-time action recognition by wearable sensors depends on quasi-periodic characteristics [21]. Furthermore, Guo et al. proposed SegrDTW, a method for rapid detection of missing or false transactions in highway ETC systems [22].

### 2.3. Time Series Data Analysis with Automatic Segmentation

We must accurately separate acceleration data for the forefist punch into parts of the forefist punch and any pre-actions in order to detect the presence or absence of pre-action. Much research has been conducted on automatic segmentation of time series data to understand motion patterns and detect change points where gestures switch. Darkhovsky et al. introduced the concept of complexity of a continuous function defined on an interval and detected and segmented change points in arbitrary nature time series data by characterizing classes of Hölder continuous functions [23]. Inoue et al. proposed an unsupervised layered segmentation that can be used as a pre-processing step for annotating time series data consisting of several types of gestures [24]. Moreover, Vögele et al. proposed an efficient method for fully automatic temporal segmentation of human behavior [25]. Wang et al. utilized automatic segmentation techniques to develop an efficient and accurate method for behavior retrieval in real-time 3D animation [26]. They have made it possible to retrieve the relevant motion in real time by matching features for the whole body and five body parts individually. In addition, automatic segmentation techniques have been applied in pHMM (pattern-based hidden Markov model), which is an algorithm for detecting correlations in time series data [27], and in DeepSense, which is a classification model for time series data from multiple sensors [28].

We may be able to classify the acceleration data of the forefist punch parts and pre-action parts with automatic segmentation techniques. However, automatic segmentation is only a technique for detecting the switch between gestures. Therefore, it is difficult to detect a pre-action, which is a very small action compared to the forefist punch. However, it is difficult to detect the pre-action that occurs during a kumite because automatic segmentation is only a technique for detecting the change between gestures and changes in a situation.

### 2.4. Detection of Motion Occurrence Timing

To detect the pre-action that occurs during a forefist punch, a method for identifying the timing of the occurrence of the pre-action can be considered. In a study to detect the start of a gesture, Sideridis et al. proposed Gesture-Keeper, a hand gesture recognition system based on RQA (recursive quantification analysis) and SVM [29]. The evaluation results showed that it correctly identified the start of a gesture with an accuracy of 87% and classified the type of gesture with an accuracy of 96% for 12 hand gestures. Yamada et al. developed a method for estimating the moment of acquiring a card in competitive karuta for contestants wearing an accelerometer and a gyroscope on their wrists [30]. They also improved the method for estimating the time of acquiring cards and applied it to actions other than competitive karuta, detecting the timing of the occurrence of arbitrary actions during various gestures [31]. They estimated the release points of three different actions: a baseball throw, a basketball free throw, and a darts throw, using an IMU attached to the wrist. The percentage of release point estimation errors of ±12 ms or less was 100% for baseball, 87.6% for basketball, and 91.1% for darts.

By applying these methods, we may be able to detect the timing of punches and kicks during kumite. However, it is difficult to apply existing methods of detecting the timing of motion occurrence to pre-actions. This is because pre-actions are not specific, fixed actions, and the waveform of acceleration data changes each time. In this study, we detected the timing of the forefist punch by applying DTW and estimated the presence or absence of a pre-action by tracing back the time series of the acceleration data and comparing it with a dataset without pre-actions.

## 3. Pre-Action Estimation Method

We propose a method to estimate the presence or absence of a pre-action by analyzing the acceleration data of karate punches to develop a practice support system to reduce pre-actions.

### 3.1. Overview

In this study, we focus on the forefist punch, the most basic kumite technique, explained in Figure 1.

The forefist punch is a technique of punching with the hand on the same side as the foot that is in front of the stance. In this study, we define “striking timing” as when the arm is fully extended and the fist reaches the target point. It is difficult to detect a pre-action in the forefist punch using the conventional gesture recognition method based on the accelerometer and the gyroscope because a pre-action has the following three features. First, the waveform of pre-action acceleration data is smaller than other gestures such as punch and kick. This feature makes it difficult to detect only the pre-action part in a series of gestures. Next, as described in Section 2.3 and Section 2.4, the pre-action is included during the gesture; therefore, it is difficult to detect the pre-action using methods that detect the point at which the gesture switches in a continuous motion, such as automatic segmentation [23,24,25,26,27,28] or detection of motion occurrence timing [29,30,31]. Finally, the waveform of pre-action acceleration data in the forefist punch is not well-defined, as there are various motions such as fist pulling, arm lowering, shoulder raising, etc., so it is difficult to construct a dataset labeled for each type of pre-action.

Therefore, we propose a method for estimating the presence or absence of a pre-action in the forefist punch by taking these features into account. An overview of the proposed method is shown in Figure 2.

The proposed method requires the preparation of a dataset consisting of only forefist punch acceleration data without pre-actions (hereinafter referred to as “dataset without pre-actions”). For any forefist punch acceleration data in which we want to recognize the presence or absence of a pre-action (hereinafter referred to as “input data”). Let μ and ν be the index of the striking timing in time series data of input data and data without pre-action in Figure 2. Moreover, let $t_{1}$ and $t_{2}$ be the length of the time series data of input data and data without pre-action. The gray lines in 3. Calculating Similarity in Figure 2 are the lines connecting the points with the smallest DTW distance at each point of the two time series data. We cut out the input data from the start of the pre-action to the striking timing in order to analyze the acceleration data that focus only on the pre-action part. Moreover, we reverse the time series of the cut input data and calculate the similarity between the input data and the dataset without pre-actions, from the striking timing to the start of the pre-action. This is because the waveform of acceleration data just before the striking timing is similar regardless of the presence or absence of a pre-action. Therefore, if there is a waveform not included in the dataset without pre-actions after the similar waveform part of the input data, we consider the waveform to be the pre-action. Section 3.2 describes the dataset without pre-actions, and Section 3.3 describes the details of the proposed method. Section 3.4 evaluates the estimation accuracy of the proposed method.

### 3.2. Dataset without Pre-Actions

As a preliminary preparation for estimating the presence or absence of pre-actions in an arbitrary forefist punch, we prepared a dataset without pre-actions consisting only of acceleration data from the forefist punch without pre-actions. We created the dataset without pre-actions by collecting the acceleration data of 30 forefist punches performed by seven karate experts (four males and three females in their 20 s) who had been practicing karate for more than seven years. We measured the acceleration data with an accelerometer attached to a stretchable wristband on the wrist of the side performing the forefist punch, as shown in Figure 3.

We used a compact wireless hybrid sensor II (WAA-010) from ATR-Promotions (Kyoto, Japan) for the measurements. The three axes of X, Y, and Z in the accelerometer are the directions shown in Figure 3. We used a ThinkPad X13 Gen 2 PC (OS: Windows 10 Home, CPU: 11th Gen Intel® Core™ i5-1135G7 @ 2.40 GHz, RAM: 16.0 GB) from Lenovo (Beijing, China). We obtained the acceleration data using the AccelViewerHybrid-II (ver.2.4.0) WAA-010 dedicated data receiving software. We generated one punch data series per participant by averaging data from 30 punches. This is because every punch, no matter how skilled, may include a small pre-action. Therefore, we constructed the dataset without pre-actions with seven punches, whose data consisted of one punch by seven separate experts.

### 3.3. Estimation Method for the Presence or Absence of Pre-Actions

We propose a method for estimating the presence or absence of pre-actions based on the similarity measured from the DTW distance between the input data and the dataset without pre-actions. Let the input data point be an arbitrary forefist punch for which we want to recognize the presence or absence of a pre-action. We denote the dataset without pre-actions as $W^{\mathrm{set}}$, the data without pre-actions as $W_{j}$, and the input data as *D*, as follows:(1)$$\begin{array}{ccc} W^{\mathrm{set}} & = & \left\{W_{j}|1\leq j\leq J\right\} \end{array}$$(2)$$\begin{array}{ccc} W_{j} & = & \left\{\left(\begin{array}{c} x_{h}\\ y_{h}\\ z_{h} \end{array}\right)|1\leq h\leq H\right\} \end{array}$$(3)$$\begin{array}{ccc} D & = & \left\{\left(\begin{array}{c} x_{i}\\ y_{i}\\ z_{i} \end{array}\right)|1\leq i\leq I\right\}. \end{array}$$

Let *J* be the number of the data without pre-actions $W_{j}$ that comprise the dataset without pre-actions $W^{\mathrm{set}}$. Furthermore, let *H* and *I* be the number of acceleration data in the three axes of the data without pre-actions $W_{j}$ and the input data *D*, respectively. We propose a method for estimating whether the input data *D* contain pre-actions, shown in Figure 4. The proposed method is based on the DTW distance between the input data *D* and the dataset without pre-actions $W^{\mathrm{set}}$, which is the reversed time series from the striking timing to the start of the pre-action, as described in Section 3.1.

We cut out the data from the start of the pre-action to the striking timing in order to analyze the acceleration data, focusing only on the pre-action part. Any punch can obtain a negative peak value in the acceleration data on the X-axis just before the striking timing. This is because the velocity of the punch rapidly approaches zero just before the striking timing in order to make the punch more powerful. In order to detect the striking timing, we obtain indices μ and $\nu_{j}$, which are, respectively, the index number at the negative peak in the acceleration data on the X-axis $x_{i}$ of the input data *D* and the X-axis $x_{h}$ of the data without pre-actions $W_{j}$. We obtain an array $W_{j}^{\prime }$ composed of the acceleration data in the three axes of the data without pre-actions $W_{j}$ that were cut out from the start of the pre-action to the striking timing. Moreover, we obtain an array $W^{\prime \;\mathrm{rev}}$ that reverses the time series of the array $W_{j}^{\prime }$. Then, we obtain an array $D^{\prime }$ composed of the acceleration data in the three axes of the cut out input data *D* from an arbitrary point *i* to the striking timing. Likewise, we obtain an array $D^{\prime \;\mathrm{rev}}$ that reverses the time series of the array $D^{\prime }$. The waveform of acceleration data just before the negative peak in the acceleration data of the punch is similar regardless of the presence or absence of a pre-action. Therefore, we can estimate the presence or absence of a pre-action based on whether the waveform that corresponds to a pre-action exists after a similar waveform.

In order to detect the waveform that corresponds to a pre-action, we calculate the DTW distance between the data without pre-actions $W^{\prime \;\mathrm{rev}}$ and the input data $D^{\prime \;\mathrm{rev}}$ in the following four steps. First, we fix the time length of the data without pre-actions $W^{\prime \;\mathrm{rev}}$ and gradually increase the time length of the input data $D^{\prime \;\mathrm{rev}}$ in the direction from the striking timing to the start of the pre-action. Next, we calculate the DTW distance in each time length of the input data $D^{\prime \;\mathrm{rev}}$ and add the DTW distance to the $CostList_{j}$. Then, we calculate the DTW distance in the three axes of X, Y, and Z at each time of the waveform using the algorithm by Myers et al. [4] and telescope the time of the waveform so that the total DTW distance is lowest. Then, we analyze a graph of the $CostList_{j}$. The difference in the $CostList_{j}$ depending on whether the input data $D^{\prime \;\mathrm{rev}}$ contain the waveform that corresponds to a pre-action is shown in Figure 5.

Consider the case when the input data $D^{\prime \;\mathrm{rev}}$ contain a pre-action, as shown in the left part of Figure 5. When comparing similar waveform parts just before the striking timing, the DTW distance decreases as the input data $D^{\prime \;\mathrm{rev}}$ become longer. However, when the input data $D^{\prime \;\mathrm{rev}}$ are lengthened to include the waveform that corresponds to a pre-action, a similarity between the data without pre-actions $W^{\prime \;\mathrm{rev}}$ and the input data $D^{\prime \;\mathrm{rev}}$ decreases. Therefore, the DTW distance increases, and the positive peak of the waveform appears in the graph of $CostList_{j}$. Then, consider the case when the input data $D^{\prime \;\mathrm{rev}}$ do not contain pre-actions, as shown in the right part of Figure 5. The positive peak value of the waveform in the graph of $CostList_{j}$ becomes smaller because there are few waveforms that correspond to pre-actions. Finally, we obtain the difference between the first local minimum and the first local maximum of the waveform in the graph of $CostList_{j}$. Let φ be the index of the first local maximum and ψ be the index of the local minimum in $CostList_{j}$. If this gap is large enough, we can consider that the input data $D^{\prime \;\mathrm{rev}}$ contain pre-actions.

Additionally, we apply the same process to all data without pre-actions $W_{j}$ from the 1st to the *J*th in the dataset without pre-actions $W^{\;\mathrm{set}}$. The data without pre-actions $W_{j}$ with the smallest local minimum can be said to be the most similar to the input data *D* in terms of motion just before the striking timing. Let $C^{\;\max}$ be the local maximum $CostList_{j}\left[\varphi \right]$ and $C^{\;\min}$ be the local minimum $CostList_{j}\left[\psi \right]$ at the data without pre-actions $W_{j}$, with the smallest local minimum in the dataset without pre-actions $W^{\;\mathrm{set}}$. We obtain the gap between $C^{\;\max}$ and $C^{\;\min}$. Then, we can estimate the presence or absence of pre-actions by whether the gap is larger than the threshold *T*. If the gap is larger than the threshold *T*, we estimate that there is a pre-action in the input data *D*. By contrast, if the gap is smaller than the threshold *T*, we estimate that there is no pre-action in the input data *D*. The threshold *T* is determined based on the experimental results in Section 3.4.

The graphs of acceleration in the three axes and DTW distance trends, when the proposed method is applied to a forefist punch without a pre-action and a forefist punch with the three types of pre-action, are shown in Figure 6. Figure 6a shows the case of a forefist punch without a pre-action. The types of pre-actions are Figure 6b pre-action of pulling the fist, Figure 6c pre-action of lowering the arm, and Figure 6d pre-action of shaking the fist sideways.

In addition, we consider the time period between the local minimum and local maximum to be the time period containing pre-actions. Therefore, we can know when the pre-action occurred in a series of forefist punches using the proposed method.

### 3.4. Experiments to Evaluate Estimation Accuracy

We conducted an experiment to verify whether the proposed method can detect pre-actions in the following three steps. First, we compared the value of the gap calculated in Section 3.3 for each presence and absence of a pre-action in the input data. Next, we determined the threshold *T* in Section 3.3 based on the result of the first step. Finally, we evaluated the accuracy of estimating the presence or absence of a pre-action while using the proposed method.

#### 3.4.1. Experiment Details

The participants were 10 karate players (eight males and two females in their 20 s) belonging to a university karate club. We divided them into two groups of those with a pre-action and those without a pre-action. Three expert karate players judged the presence or absence of a pre-action in the participants. The environment for measuring the acceleration data was the same as when creating the dataset without pre-actions described in Section 3.2. First, to verify whether the proposed method can detect a pre-action, we attached an accelerometer to the wrist of each participant performing a forefist punch and measured forefist punching data 10 times. We calculated the average value of the gap using the proposed method, using the 10 punches’ worth of data measured for each participant as input data. We investigated whether the average value of the gap differed depending on the presence or absence of pre-actions by the participant.

Next, to determine the threshold *T*, we set the five thresholds shown in Table 1 based on the results of the above investigation and evaluated the accuracy of estimating the presence or absence of pre-actions at each threshold value. We estimated the presence or absence of pre-actions using the proposed method for 10 punches’ worth of data measured for each participant. We evaluated the estimation accuracy for each of the five threshold values and defined the threshold with the highest accuracy as the threshold *T* used in this study.

Finally, to investigate the effectiveness of the proposed method, we compared the estimation accuracy of the baseline method and the proposed method. Here, the baseline method estimates the presence or absence of pre-actions by measuring the similarity only by the sum of the DTW distances for each of the three axes in the input data and comparison data. A comparison of the methods for estimating the presence or absence of pre-actions between the proposed method and the baseline method is shown in Figure 7 and Figure 8.

In the baseline method, we calculated the sum of the DTW distances in the three axes across the whole dataset from the beginning of the punch to the end of the punch for each punch data series, without any processing, such as cut back to the striking timing or time series reversal. When the punch data from one of the 10 participants were used as input data, the comparison data were the punch data of the remaining nine participants. We measured the DTW distance between the input data and the punch data of the other nine participants. Then, we estimated the presence or absence of pre-actions by reference to the participant group with the lowest DTW distance of the nine participants. When the *m*th (1≤n≤10) punch $D_{m}$ by the *n*th (1≤m≤10) participant is taken as input data, we calculate the average of the DTW distance with the 10 punches’ worth of data by each of the remaining nine participants. The method for estimating the participant who has the lowest average among them is shown in Figure 9. Let r(1≤r≤10,r≠n) denote any participant index among the nine participants and l(1≤l≤10) denote any punch’s data index among the 10 punches’ worth of data of the *r*th participant.

#### 3.4.2. Results and Discussion

The average value of the gap obtained for each participant is shown in Figure 10a.

The experimental results show that participants with a pre-action have a larger gap, and participants without a pre-action have a smaller gap. Individual differences between participants with pre-actions were large. We compared the acceleration data in the X, Y, and Z-axes for each participant and found that the waveform of the acceleration data in the X-axis was similar regardless of the presence or absence of a pre-action. Participants with higher average values for the gap showed more noise in the waveform of pre-actions on the Y-axis and Z-axis. As shown in Figure 10b, we conducted a *t*-test for the mean difference between the overall local maximum and local minimum in the presence and absence of pre-actions and found a significant difference (*t*(5) = 3.7749, *p*< 0.05).

Next, the result of estimating the presence or absence of a pre-action using the proposed method is shown in Figure 11. The results show that the precision increases and the recall decreases as the threshold value increases. The threshold value β gave the best overall accuracy, with 86.1% accuracy, 84.7% precision, 85.6% recall, and 85.2% F-measure. Therefore, the threshold in the proposed method is set to $T=0.90\times 10^{8}$.

Finally, a comparison of the estimation accuracy between the baseline method and the proposed method is shown in Table 2. The baseline method had a 46.0% accuracy, 46.4% precision, 52.0% recall, and 49.1% F-measure. The proposed method improved the accuracy by 40.1% and the F-measure by 36.1% compared to the baseline method. Hence, we found that the proposed method is effective for estimating the presence or absence of a pre-action.

## 4. Practice Support System

We developed a system to support practice to reduce pre-actions called KARATECH using the proposed method described in Section 3. An overview of KARATECH is shown in Figure 12. As in Section 3.4, the user wears an accelerometer on the wrist that performs a forefist punch. When the system starts, it measures the acceleration data during the forefist punch and estimates the presence or absence of a pre-action using the proposed method. KARATECH shows the user the result in the video with the pre-action part slowed down, and the graph of the acceleration data of each axis is compared to the data without pre-actions. The user practices with reference to the feedback results and repeats the measurement. This allows the user to intuitively understand their pre-actions and to practice more efficiently. Section 4.1 describes the requirements for designing KARATECH, Section 4.2 describes the implementation method, and Section 4.3 describes the procedure for using KARATECH. Section 4.4 evaluates the effectiveness of KARATECH in reducing pre-actions.

### 4.1. System Design

In the development of a system to support the practice of reducing pre-actions in the forefist punch, the following requirements were identified as necessary: the system should be able to measure their motions, intuitively understand their pre-actions, and maintain the accuracy of the forefist punch. Therefore, KARATECH has the following functions: acceleration measurement, video recording/playback, graph display, and velocity display. In order to estimate whether the user’s forefist punch includes a pre-action or not, it is necessary to measure acceleration during the forefist punch. The system uses the proposed method to estimate the presence or absence of a pre-action in the measured data and shows the results through video playback and the graph of acceleration data. We elaborate on the feedback of results. First, because a pre-action in kumite techniques is momentary, it is difficult to observe through mirrors, etc. Therefore, it is necessary to provide feedback on the motions that correspond to pre-actions in real-time and in an easy-to-understand manner. If there is a pre-action in the input data, the system calculates the time period corresponding to the pre-action using the proposed method and plays back a slowed-down video of that part. The user can easily understand what kind of motions are included in their pre-action. Next, we found from Section 3.4 that there is a large difference in the waveform of acceleration data on the Y-axis and Z-axis of the forefist punch between the presence and absence of a pre-action. Therefore, it is necessary to understand which axis in particular is causing the pre-action in the user’s motion. KARATECH draws the graph comparing the acceleration data in the three axes of the input data and the dataset without pre-actions. The displayed graph allows the user to check which axis of the dataset is causing the most unnecessary motions. In addition, when practicing with KARATECH, it is necessary to be careful not to reduce the accuracy of the forefist punch by focusing too much on reducing the pre-action. Therefore, the system uses the velocity of the forefist punch as an indicator of the accuracy of the forefist punch. KARATECH displays the estimated velocity calculated by the trapezoidal integration method after removing the effects of noise and gravitational acceleration. The user uses the displayed velocity as a reference to maintain the accuracy of the forefist punch.

The user interface of KARATECH is designed to satisfy the requirements described in Figure 13. In order to satisfy these requirements, an accelerometer capable of measuring acceleration and a web camera capable of capturing the user’s motions are required.

### 4.2. System Implementation

The equipment used for the implementation of KARATECH is as follows. As in Section 3.2, we used compact wireless hybrid sensor II (WAA-010) from ATR-Promotions (Kyoto, Japan) as the accelerometer for measuring acceleration. We used a ThinkPad X13 Gen 2 PC (OS: Windows 10 Home, CPU: 11th Gen Intel® Core™ i5-1135G7 @ 2.40 GHz, RAM: 16.0 GB) from Lenovo (Beijing, China), and the communication method between the accelerometer and the PC was Bluetooth communication. We used a C505 webcam from Logicool (Tokyo, Japan) to capture the user’s motions.

### 4.3. Usage Procedure

The procedure for practicing with the system that satisfies the above requirements is described below.

Camera SelectionThe camera devices recognized by the PC are displayed in a pull-down list. The user selects the camera to be used.Sensor ConnectionThe user clicks on the “Connect” button after wearing an accelerometer on the arm and turning on the power to connect the PC and the accelerometer. After that, the camera is activated, and the image is displayed on the screen.Motion MeasurementA five-second countdown begins after the user clicks the “Measure” button. During this time, the user performs a forefist punch in the camera position.Pre-action EstimationThe system estimates the presence or absence of pre-actions using the method described in Section 3.3 referring to the measured data.Measurement Results DisplayThe system displays the presence or absence of pre-actions, the video with the pre-action part slowed down, and the graph compared to the dataset without pre-actions after the user clicks the “Play” button. Furthermore, the system also displays the estimated velocity of the forefist punch, which the user can use as an indicator to maintain the accuracy of the forefist punch. The user can view the video repeatedly by pressing the “Play” button again.Repeat PracticeThe user clicks the “Measure” button again to practice the forefist punch, referring to the displayed results. By repeating the process from 3. Motion Measurement to 5. Measurement Results Display, the user can deepen their understanding of their pre-actions and practice more efficiently.

### 4.4. Experiments to Evaluate the Effectiveness of Pre-Action Reduction

In order to evaluate the effectiveness of using KARATECH for reducing pre-actions, we conducted a short-term experiment and a long-term experiment. In this section, we first describe the short-term experiment, then explain the functional modifications of KARATECH, and finally describe the long-term experiment.

#### 4.4.1. Short-Term Experiment

We conducted a short-term experiment to test whether a one-off practice using the proposed system would immediately reduce pre-actions and verify appropriate feedback methods. The participants were four intermediate karate players (two teenage males and two teenage females) belonging to a university karate club and eight beginner karate players (eight males in their 20 s), for a total of 12. The participants were different from the participants in Section 3.4. The experimental procedure is shown in Figure 14.

Let the pre-action rate be defined as the rate of pre-actions among the forefist punches performed an arbitrary number of times. First, we measured the acceleration data in the forefist punch 50 times in order to investigate the pre-action rate of the participants before practice. Next, we divided the participants into two groups: one group practiced using KARATECH (hereinafter referred to as “the system-using group”), and the other group practiced by themselves while looking only at a mirror (hereinafter referred to as “the non-using group”). The practice time was 30 min. Finally, we measured the acceleration data of 50 forefist punches again to investigate how much the pre-action rate had decreased compared to the first measurement. We estimated the pre-action rate using the proposed method described in Section 3.

We experimented in the environment shown in Figure 15. In the environment of the system-using group (upper portion of Figure 15), there was a PC for controlling KARATECH, a display for showing the system user interface, and a webcam for recording the participant’s motions. Similarly, in the experiment conducted in Section 3.4, the participant wears an accelerometer on the wrist of the side performing the forefist punch, and a mouse is placed near the participant to control the system. The non-using group (lower part of Figure 15) practices while looking only at a mirror. This experiment was conducted with the approval of the Ethical Review Committee for Research Directly Involving Human Participants of the Graduate School of Engineering, Kobe University (approval numbers 04–06).

#### Results of Short-Term Experiment

The averages of the estimated velocities of 50 forefist punches measured before and after practice for each participant are shown in Table 3. In this experiment, the velocity of the forefist punch was displayed as one of the indicators to maintain the accuracy of the forefist punch. Table 3 shows that the velocity of the forefist punch increased after practice compared to before practice for most participants. This result shows that the participants could practice without losing the accuracy of the forefist punch.

The differences between the pre-action rates in 50 forefist punches measured before and after practice are shown in Figure 16. We conducted a two-way between-participants ANOVA to analyze the differences between pre-action rates in 50 forefist punches measured before and after practice, using two factors: whether or not KARATECH was used during practice and karate proficiency level. The results showed no significant main effect due to the usage of KARATECH, but a significant main effect was found for karate proficiency level (F(1,8) = 5.39, *p* < 0.05). When we conducted tests for simple main effects, significant simple main effects were found for the presence or absence of KARATECH use among beginners (F(1,8) = 8.42, *p* < 0.05) and for the level of proficiency in participants who did not use KARATECH (F(1,8) = 12.32, *p* < 0.01).

#### Discussion of Short-Term Experiment

We found that the effect of reducing pre-actions was high for beginners and low for intermediate players by one-off practice using KARATECH. All beginners were able to perform a powerful forefist punch after practice. All beginners in the system-using group showed a decrease in pre-actions after practice, whereas all beginners in the non-using group showed an increase in pre-actions after practice. The beginners in the system-using group could practice the forefist punch with fewer pre-actions because they could recognize their pre-actions during practice. In contrast, the effect of reducing pre-actions was low for all intermediate players. Because it was difficult to reduce pre-actions in a short time, intermediate players had already formed their forefist punch form through practicing for a long time. Therefore, we found that KARATECH was highly effective in reducing pre-actions even when used in one-off practice with beginners in karate.

#### 4.4.2. Functional Modifications to KARATECH

Based on the results of the questionnaire for the short-term experimental participants in Section 4.4.1, we modified some of the feedback methods of KARATECH. The user interface of the modified KARATECH is shown in Figure 17.

We removed the feedback of graphs of acceleration data and added feedback of diagrams and scores. We had displayed the graph of acceleration data to show which axis had the largest pre-action. However, some participants who do not usually use graphs preferred a more intuitive feedback method. Therefore, we added a feedback method that displays a diagram of the axis with the largest pre-action. Furthermore, if the acceleration data of the forefist punch include pre-actions, a “pre-action level” is displayed to show how large the pre-action was. If the acceleration data of the forefist punch do not include pre-action, a “score” is displayed to show how small the pre-action was. We calculated these regarding the gap and the threshold *T* in Section 3.

#### 4.4.3. Long-Term Experiment

We conducted a long-term experiment to verify the reduction effect on the pre-action rate when practicing continuously with KARATECH. The participants were six intermediate karate players (two teenage males and two teenage females) belonging to a university karate club and eight beginner karate players (eight males in their 20 s), for a total of 14. The experimental schedule is shown in Figure 18.

First, we conducted a test to measure the pre-action rate of the participants before practice. Next, the participants practiced twice a week and took a test once a week for four weeks. Finally, one month after the fourth-week test, we conducted a test to investigate the retention of the effect of reducing the pre-action rate. As in Section 4.4.1, we divided the participants into the system-using group and the non-using group, and each group practiced for 15 min to reduce the pre-action rate of the forefist punch. During the test, the participants wore an accelerometer on their arms to measure the acceleration data of 50 forefist punches. We also recorded the test with a video camera. In the long-term experiment, we evaluated the pre-action rate during the test by the method described in Section 3 and by video evaluation with three karate experts. The experimental environment was the same as in Section 4.4.1. This experiment was conducted with the approval of the Ethical Review Committee for Research Directly Involving Human Participants of the Graduate School of Engineering, Kobe University (approval numbers 05–11).

#### Results of Long-Term Experiment

Before the results of the experiment, we note that we analyzed the results using data from two intermediate participants in the system-using group and three participants in the non-using group because one intermediate participant in the system-using group was unable to continue the experiment due to injury. The averages of the forefist punch velocity for each participant during each test, depending on whether KARATECH was used or not and proficiency level, are shown in Figure 19. In both groups, the velocity of the forefist punch was not significantly changed by the long-term practice for reducing pre-actions.

The differences in the pre-action rate between Test 0 and Test 4 by the evaluation of the proposed method and the video evaluation with the karate experts are shown in Figure 20. We conducted a two-way between-participants ANOVA to analyze the changes in pre-action rates from Test 0 to Test 4, as determined by both evaluation methods, using two factors: whether or not KARATECH was used during practice and the karate proficiency level. The results showed no significant differences due to the usage of KARATECH, proficiency level, or their interaction for either evaluation method.

The pre-action rate for each test in each participant group evaluated by the proposed method is shown in Figure 21, and the rate evaluated by video evaluation with three karate experts is shown in Figure 22.

In the evaluation of the proposed method, the system-using group showed that the pre-action rate decreased with practice, regardless of proficiency level. Only one intermediate player in the non-using group had a very high pre-action rate in Test 0, and this rate decreased significantly with practice. In the video evaluation by the karate experts, the pre-action rate showed a decreasing trend regardless of whether KARATECH was used or not and regardless of the proficiency level. We evaluated the degree of retention of the pre-action reduction effect after one month of practice based on the change in the pre-action rate from Test 4 to Test 5 for both evaluation methods. Because there was no significant change in the pre-action rate, we confirmed that the pre-action reduction effect was maintained when KARATECH was used.

Questionnaire survey questions about KARATECH for the system-using group are shown in Table 4, and the results are shown in Figure 23. In the results shown, 5 represents “easy to understand” or “felt”, and 1 represents “difficult to understand” or did not feel in Figure 23. Many participants answered that the use of KARATECH, the feedback methods, the video playback, and the velocity display were easy to understand. In addition, all participants answered that KARATECH assisted them in practicing to reduce pre-actions.

#### Discussion of Long-Term Experimentsection

Based on the results of the long-term experiment, we found that the reduction in the pre-action rate was greater when KARATECH was used in both evaluation methods. The results also showed that the participants felt that KARATECH helped them to reduce their pre-actions. This indicates that KARATECH is effective in reducing pre-action for beginners and intermediate karate players. In conclusion, we found that continuous use of KARATECH was effective not only for beginners but also for intermediate karate players.

We verified the accuracy of the proposed method by comparing the pre-action rate between the proposed method evaluation and the video evaluation by karate experts. The differences in pre-action rates between the two methods for each participant are shown in Table 5. Let the difference be the value created by subtracting the pre-action rate of the video evaluation from the pre-action rate of the proposed method evaluation. Differences larger than 0.25 are shown in bold. The value that is the third quartile when all differences are arranged in ascending order of their absolute values is 0.25. The table shows that the differences were particularly large in the evaluation results of participants LD, LJ, and LN. Since these three participants were faster in the forefist punch than the other beginners, it may have been difficult for even the experts to understand the pre-action by only video evaluation.

## 5. Conclusions

In this study, we developed KARATECH, which supports practice to reduce pre-actions, an important element in karate. The beginner and intermediate karate players can intuitively understand their pre-actions and practice efficiently by using KARATECH. First, we proposed a method for estimating the presence or absence of pre-actions. The proposed method is based on the DTW distance between the acceleration data of arbitrary punches and a previously prepared dataset consisting of only acceleration data of punches without pre-actions. Next, we investigated the estimation accuracy of the proposed method. As a result, we found that the proposed method could estimate the presence or absence of pre-actions with an accuracy of 86.1%, 40.1% higher than that of the baseline method. Finally, we developed KARATECH and experimented to evaluate the effect of the system in reducing pre-actions. The results showed that continuous practice using KARATECH was more effective in reducing pre-actions than practice while looking at a mirror alone.

In the future, we plan to improve the estimation accuracy of pre-actions and investigate the accuracy of estimation in an active state. First, in order to improve the accuracy of estimating the presence or absence of pre-actions, we will expand the data registered in the dataset without pre-actions and consider the position of the accelerometer. We will investigate the change in estimation accuracy by attaching accelerometers to different parts of the body and by increasing the number of sensors. Next, it is necessary to investigate whether it is possible to accurately estimate the presence or absence of pre-actions even when the player is in an active state, such as during a kumite match, because we had estimated the presence or absence of pre-actions in a static state.

## Figures and Tables

**Figure 1 sensors-24-02306-f001:**
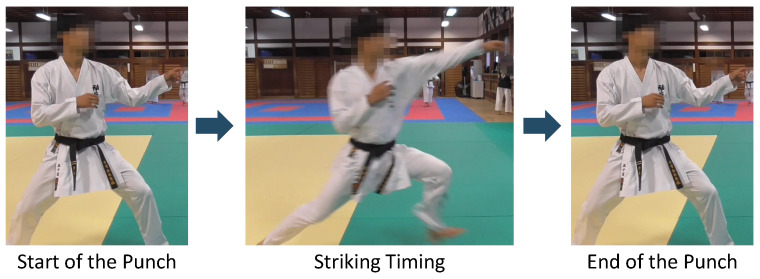
How to perform the forefist punch.

**Figure 2 sensors-24-02306-f002:**
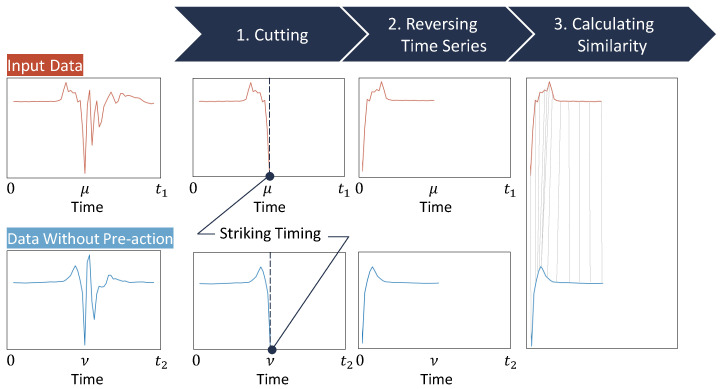
Overview of methods for estimating the presence or absence of pre-actions.

**Figure 3 sensors-24-02306-f003:**
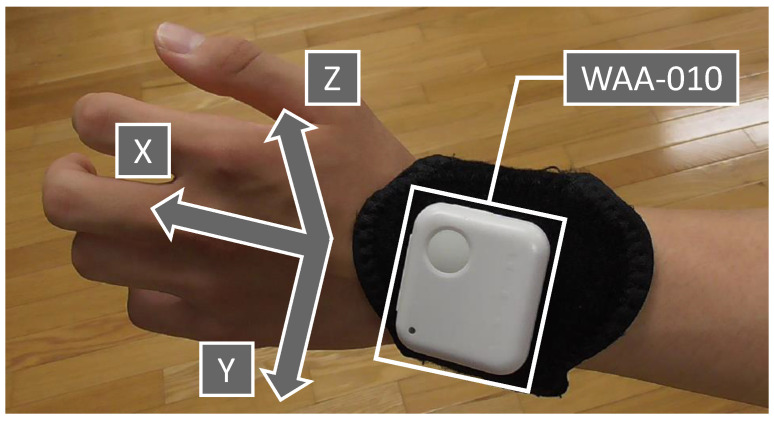
Mounting position of accelerometer (WAA-010) and direction of the three axes.

**Figure 4 sensors-24-02306-f004:**
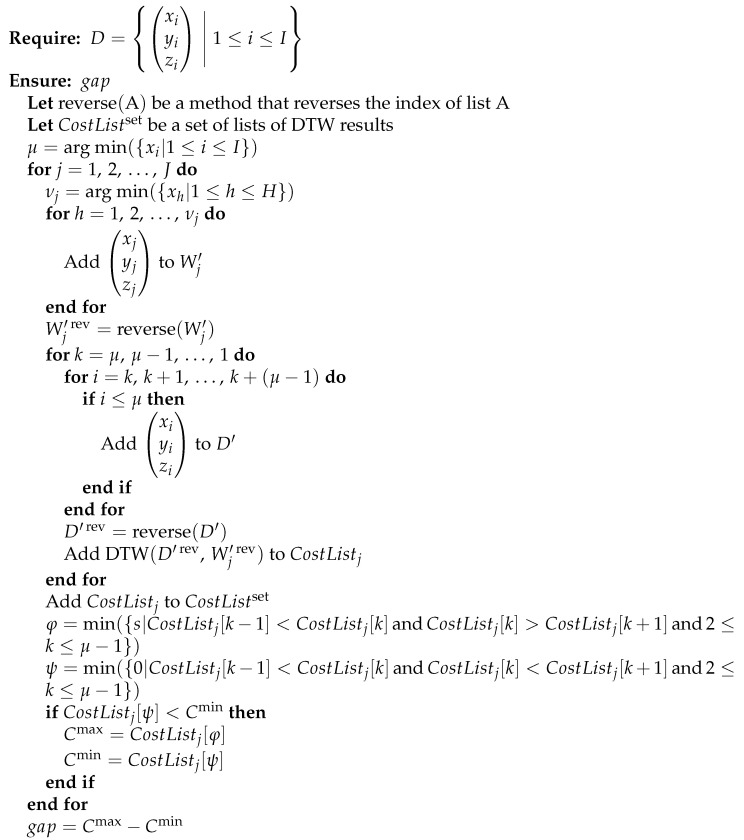
Proposed method for estimating whether an arbitrary forefist punch contains a pre-action.

**Figure 5 sensors-24-02306-f005:**
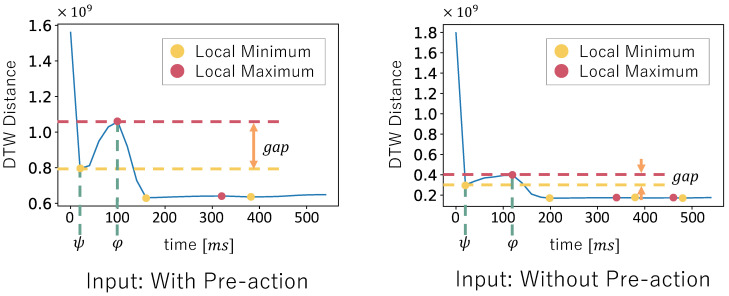
Comparison of DTW distance trend graphs in the presence and absence of pre-actions.

**Figure 6 sensors-24-02306-f006:**
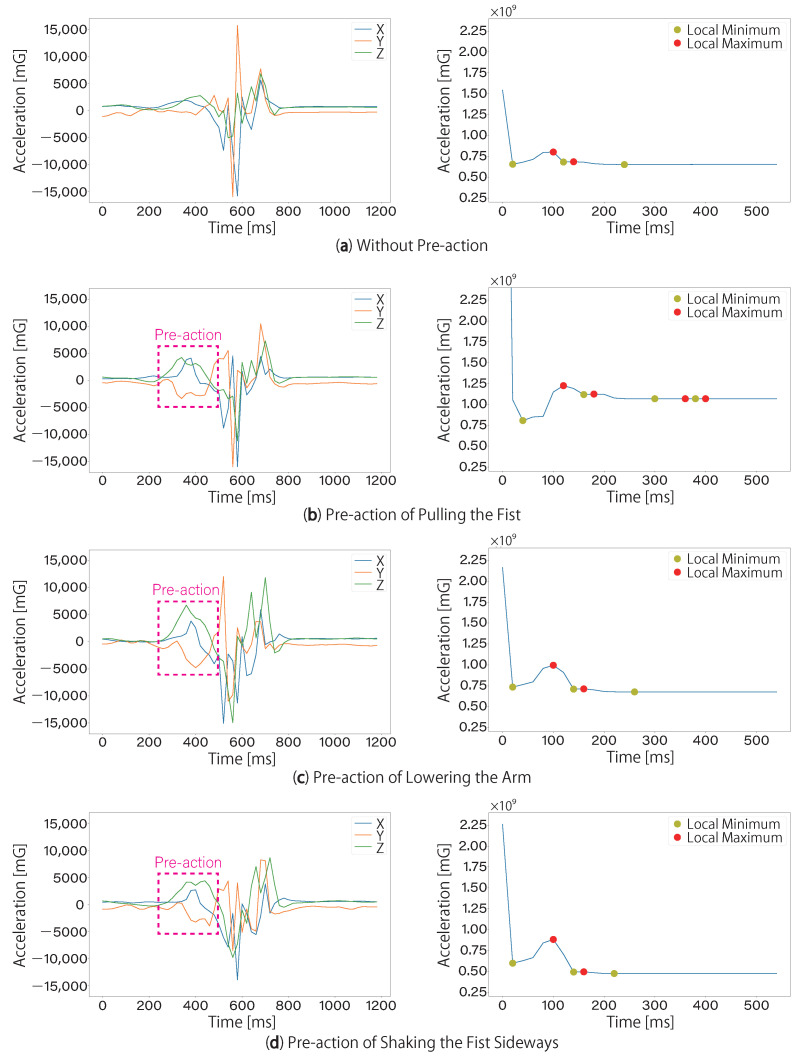
Comparison of graphs of the three axes of acceleration and DTW distance trends for forefist punch without pre-actions and forefist punch with the three types of pre-action.

**Figure 7 sensors-24-02306-f007:**
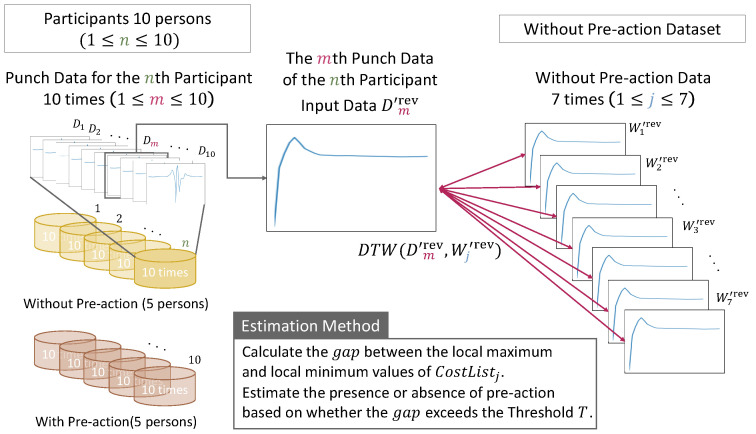
Proposed method for estimating the presence or absence of a pre-action.

**Figure 8 sensors-24-02306-f008:**
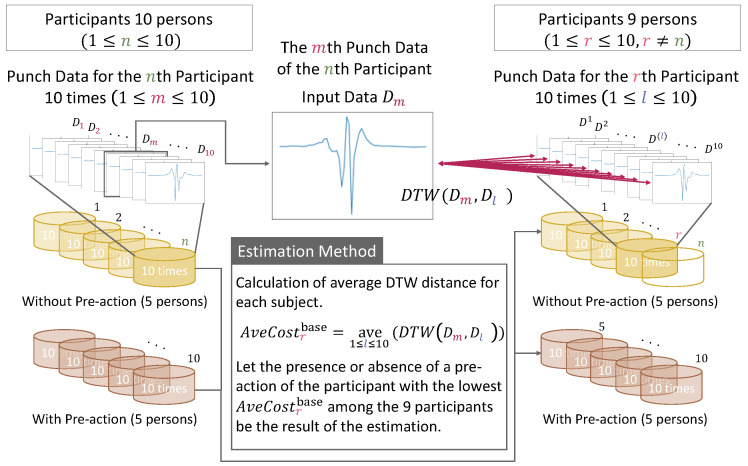
Baseline method for estimating the presence or absence of a pre-action.

**Figure 9 sensors-24-02306-f009:**
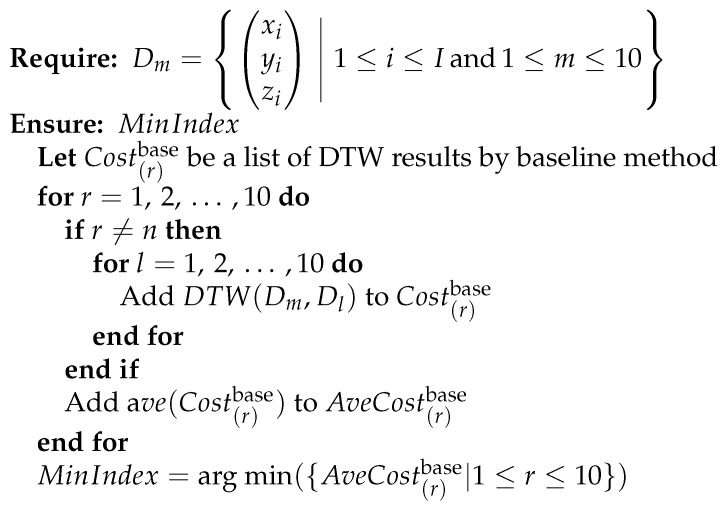
Algorithm for estimating whether an arbitrary forefist punch contains a pre-action by the baseline method.

**Figure 10 sensors-24-02306-f010:**
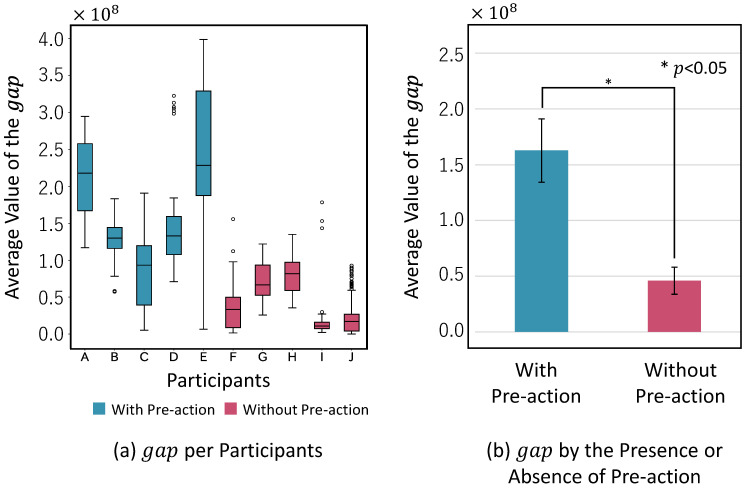
Investigation results for the gap in the presence or absence of a pre-action.

**Figure 11 sensors-24-02306-f011:**
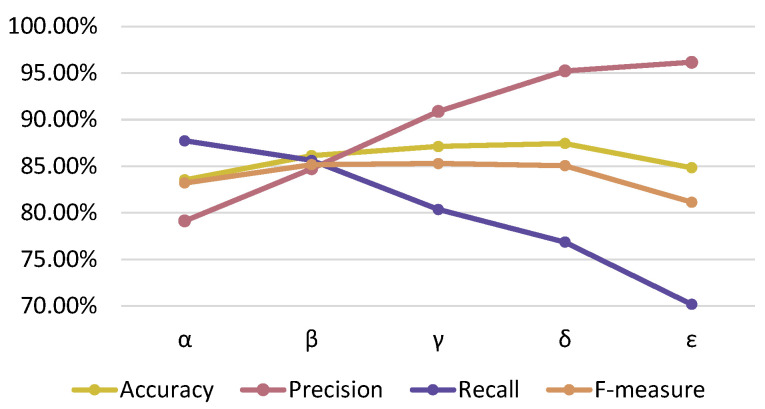
Estimation accuracy by threshold value.

**Figure 12 sensors-24-02306-f012:**
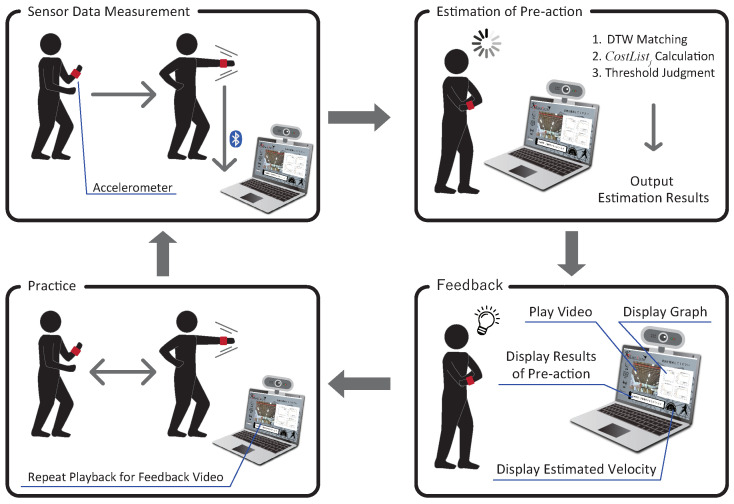
Overview of KARATECH, a system to support practices that reduce pre-actions.

**Figure 13 sensors-24-02306-f013:**
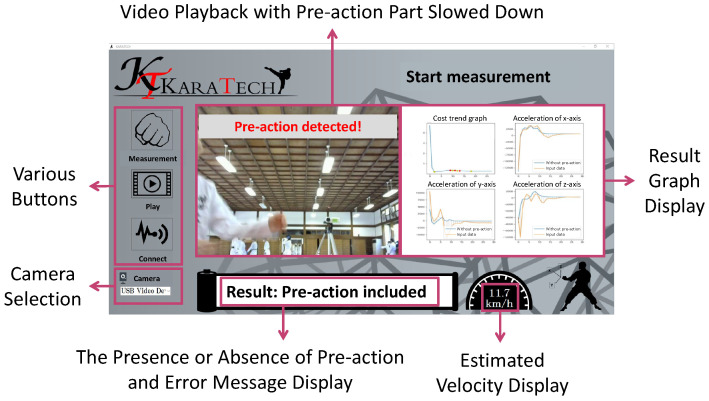
User interface of KARATECH.

**Figure 14 sensors-24-02306-f014:**
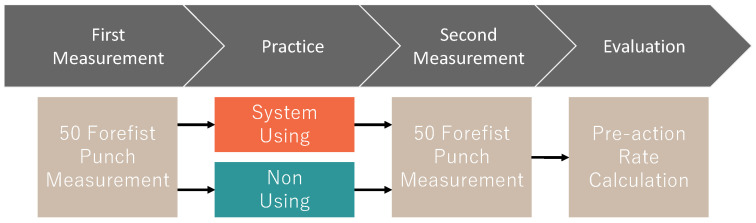
Short-term experimental procedure.

**Figure 15 sensors-24-02306-f015:**
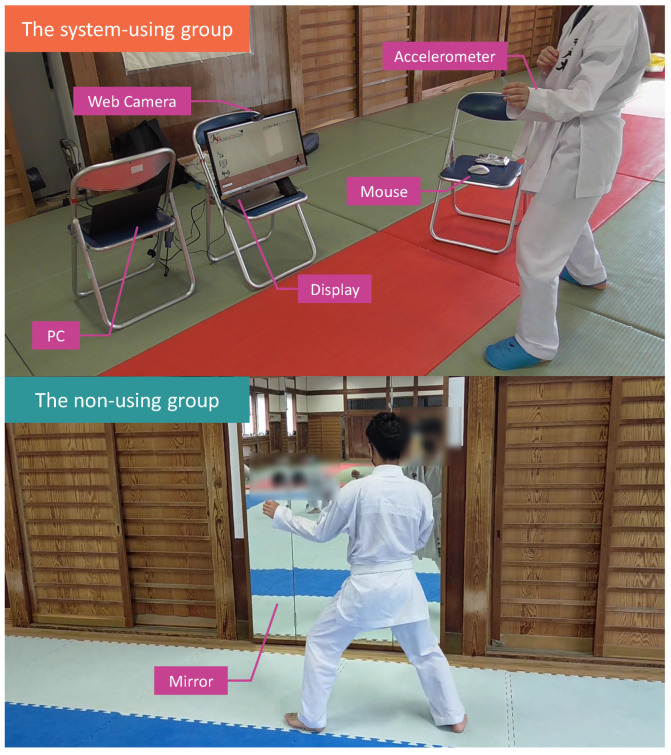
Experimental environment.

**Figure 16 sensors-24-02306-f016:**
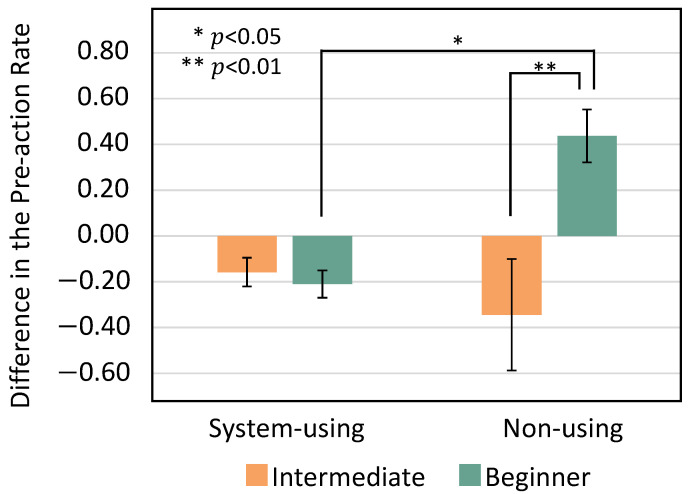
Comparison of pre-action rates before and after practice.

**Figure 17 sensors-24-02306-f017:**
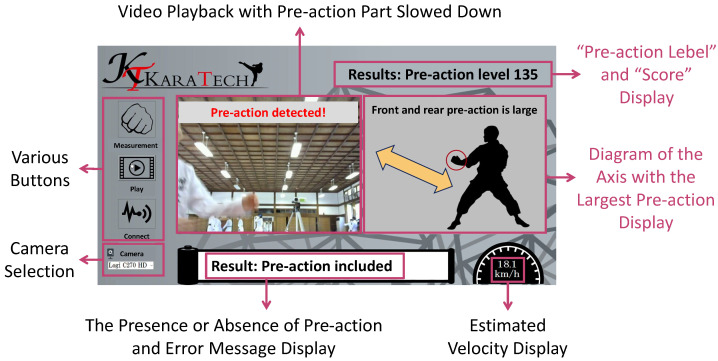
Modified KARATECH user interface.

**Figure 18 sensors-24-02306-f018:**
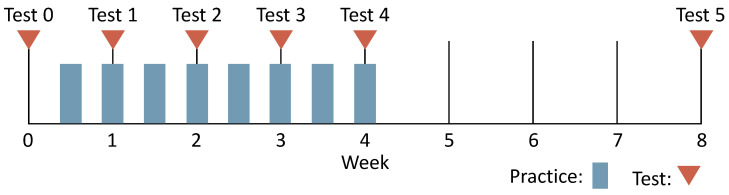
Long-term experiment schedule.

**Figure 19 sensors-24-02306-f019:**
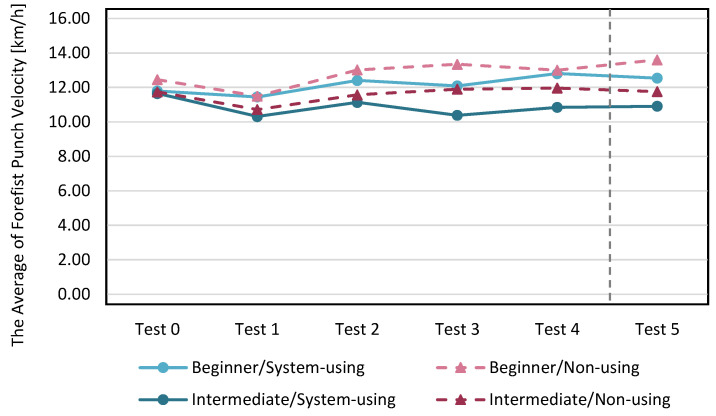
Average of forefist punch velocity per test.

**Figure 20 sensors-24-02306-f020:**
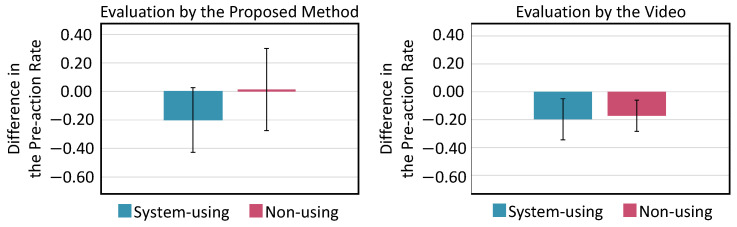
Differences in pre-action rates between Test 0 and Test 4 for each evaluation method.

**Figure 21 sensors-24-02306-f021:**
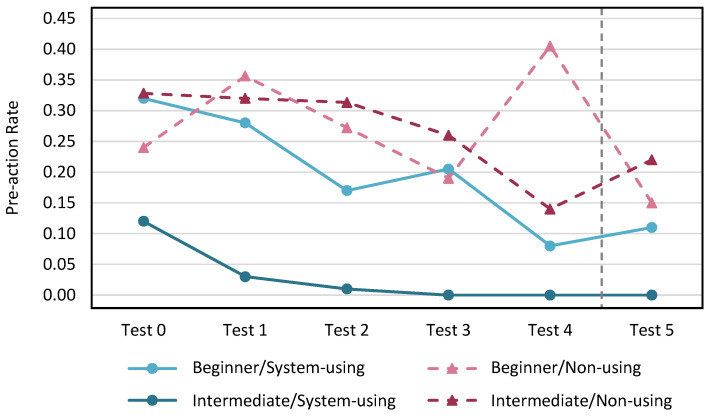
The pre-action rate per test evaluated by the proposed method.

**Figure 22 sensors-24-02306-f022:**
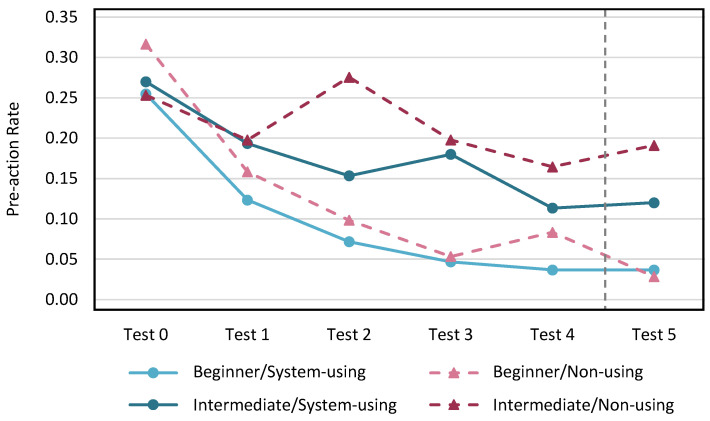
The pre-action rate per test evaluated by the video.

**Figure 23 sensors-24-02306-f023:**
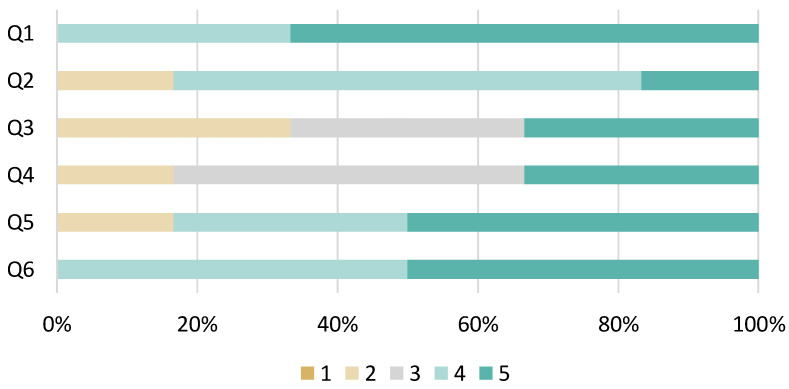
Results of questionnaire survey on KARATECH.

**Table 1 sensors-24-02306-t001:** Five threshold patterns used in the experiment.

Pattern	Threshold
α	$0.80\times 10^{8}$
β	$0.90\times 10^{8}$
γ	$1.00\times 10^{8}$
δ	$1.10\times 10^{8}$
ϵ	$1.20\times 10^{8}$

**Table 2 sensors-24-02306-t002:** Comparison of the accuracy of estimating the presence or absence of a pre-action by the baseline method and the proposed method.

	Baseline Method	Proposed Method
Accuracy	46.0%	86.1%
Precision	46.4%	84.7%
Recall	52.0%	85.6%
F-measure	49.1%	85.2%

**Table 3 sensors-24-02306-t003:** Average velocity of the forefist punch before and after practice (unit: km/h).

Participants	Group	Proficiency	Before	After
A	System-using	Intermediate	12.11	9.28
B		Intermediate	9.92	10.31
C		Beginner	13.30	14.39
D		Beginner	10.29	13.00
E		Beginner	13.21	12.28
F		Beginner	6.28	10.92
G	Non-using	Intermediate	12.89	14.43
H		Intermediate	11.31	10.51
I		Beginner	11.38	12.32
J		Beginner	10.66	11.99
K		Beginner	11.24	11.57
L		Beginner	8.75	10.85

**Table 4 sensors-24-02306-t004:** Survey questions about KARATECH.

No.	Question
Q1	Was it easy to understand how to use the system?
Q2	Was the video feedback easy to understand?
Q3	Was the graphical feedback easy to understand?
Q4	Was the feedback by score easy to understand?
Q5	Was the feedback by velocity easy to understand?
Q6	Did you feel that it helped you practice reducing pre-action?

**Table 5 sensors-24-02306-t005:** Difference in pre-action rates between proposed method evaluation and video evaluation.

Participants	Group	Proficiency	Test 0	Test 1	Test 2	Test 3	Test 4	Test 5
LA	System-using	Intermediate	−0.153	−0.220	−0.187	**−0.293**	−0.187	−0.227
LB		Intermediate	−0.147	−0.107	−0.100	−0.067	−0.040	−0.013
LC		Beginner	−0.080	−0.093	0.067	0.073	0.073	0.040
LD		Beginner	**0.400**	**0.647**	**0.387**	**0.560**	0.067	0.207
LE		Beginner	**0.427**	0.093	−0.033	0.027	0.020	0.073
LF		Beginner	**−0.487**	−0.020	−0.027	−0.027	0.013	−0.027
LG	Non-using	Intermediate	0.193	0.180	0.033	0.033	**−0.367**	−0.013
LH		Intermediate	−0.067	−0.027	−0.047	0.013	0.000	−0.007
LI		Intermediate	0.098	0.213	0.127	0.140	**0.293**	0.107
LJ		Beginner	0.240	**0.693**	**0.253**	0.013	**0.547**	0.027
LK		Beginner	**−0.353**	−0.115	0.107	0.120	**0.280**	0.067
LM		Beginner	**−0.360**	−0.067	−0.040	0.013	−0.047	−0.060
LN		Beginner	0.167	**0.280**	**0.376**	**0.400**	**0.507**	**0.453**

## Data Availability

Not published due to ethical restrictions. If you want to browse data, please contact the author personally.

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
