# Peer review of "KARATECH: A Practice Support System Using an Accelerometer to Reduce the Preliminary Actions of Karate"

_sensors, 2024, doi:10.3390/s24072306_

Round 1

Reviewer 1 Report

Comments and Suggestions for Authors

Here is the translation of your questions into English, maintaining an academic tone and adhering to formal language standards:

1. The methods discussed appear to be dated, with references several years old. Are there any research methodologies from the past three years that have been considered in this study?

2. The second paragraph lists numerous methods but does not provide a sufficient introduction to the chosen methodology. Furthermore, it fails to clarify the reasons for selecting these specific methods.

3. Please explain the rationale behind performing sequence reversal on the trimmed dataset.

Author Response

Thank you very much for providing important comments. We are thankful for the time and energy you expended. Our responses to the reviewer's comments are in the attached file.

Reviewer 2 Report

Comments and Suggestions for Authors

Authors present an interesting application of an accelerometer in the area of sports science. Authors obtained good results for the detection of pre-actions in Karate sports. The manuscript is well written and can be accepted after minor revision. This reviewer's comments are as follows.

1. The criteria of a pre-action of a forefist is not clear.

Can we say that a 5-cm movement forward/backward of a forefist in a short time (0.05 s) is a pre-action?

Or is it for a movement over more than 15 cm in a less amount of time (e.g. 0.7 s)?

2. Please include many curves of actual measured accelerometer curve with various cases of pre-action forefist movements. And analyze them in the view point of the proposed algorithm.

Author Response

(The authors gave the same response as above.)

Reviewer 3 Report

Comments and Suggestions for Authors

This paper proposes a method to estimate the presence or absence of preliminary actions based on DTW using an accelerometer as a practice support system to reduce preliminary actions in Karate.

In order to determine the presence or absence of a pre-action, the authors propose a method of determining the presence or absence of a pre-action by determining the mutual similarity between data without pre-action and input data from the start of the pre-action to the hitting timing. Since similarity determination is based on DWT, it is considered an appropriate method because it can be applied to data length or nonlinear data.

The experimental approach and results are presented in a very systematic manner as well as the interpretation, and the academic contribution is also evaluated as high. This is a well-written paper that requires no special comment.

Author Response

Thank you very much for providing important comments. We are thankful for the time and energy you expended.